

# Spatial and seasonal patterns of water isotopes in northeastern German lakes

Bernhard Aichner[1], David Dubbert[2], Christine Kiel[3], Katrin Kohnert[3], Igor Ogashawara[3], Andreas
Jechow[2,3],    Sarah-Faye Harpenslager[1], Franz Hölker[2], Jens Christian Nejstgaard[3], Hans-Peter
Grossart[3,4], Gabriel Singer[2,5] Sabine Wollrab[3], Stella A. Berger[3]

[1]  Leibniz Institute of Freshwater Ecology and Inland Fisheries, Dep. 2 Ecosystem Research,
Müggelseedamm 301, Berlin, Germany
[2] Leibniz Institute of Freshwater Ecology and Inland Fisheries, Dep. 1 Ecohydrology, Müggelseedamm
310, Berlin, Germany
[3] Leibniz Institute of Freshwater Ecology and Inland Fisheries, Dep. 3 Experimental Limnology, Zur
alten Fischerhütte 2, D-16775 Stechlin
[4] Potsdam University, Institute for Biochemistry and Biology, Maulbeerallee, D-14469 Potsdam,
Germany
[5] Department of Ecology, Innsbruck University, Technikerstrasse 25, A-6020 Innsbruck, Austria

*Correspondence to*: Bernhard Aichner (bernhard.aichner@gmx.de)

**Abstract.** Water isotopes ($\delta^2$H and $\delta^{18}$O) were analyzed in samples collected in lakes associated to

riverine systems in northeastern Germany throughout 2020. The dataset (Aichner et al., 2021) is derived

from water samples collected at a) lake shores (sampled in March and July 2020); b) buoys which were

temporarily installed in deep parts of the lake (sampled monthly from March to October 2020); c)

multiple spatially distributed spots in four selected lakes (in September 2020); d) the outflow of

Müggelsee (sampled biweekly from March 2020 to January 2021). At shores, water was sampled with

pipette from 40-60 cm below water surface and directly transferred into a measurement vial, while at

buoys a Limnos water sampler was used to obtain samples from 1 m below surface. Isotope analysis

was conducted at IGB Berlin, using a Picarro L2130-i cavity ring-down spectrometer. The data give

information about the seasonal isotope amplitude in the sampled lakes and about spatial isotope

variability in different branches of the associated riverine systems.

# 1 Introduction

The varying physical properties of stable oxygen and hydrogen isotopes of the water molecule ($^{16}$O, $^{17}$O, $^{18}$O, $^{1}$H, $^{2}$H) result in isotope fractionation during evaporation and condensation processes. This causes spatial and temporal variability of the isotopic signature of water in both the liquid and the vapor phase (Craig, 1961; Dansgaard, 1964; Gat and Gonfiantini, 1981). Water stable isotopes are therefore ideal tracers of hydrological processes, for example in lacustrine and riverine systems. Specifically in

lakes, their general morphology (water volume, lake area, depth and shape), the mean residence time of the water, the balance between in- and outflows of multiple types (surface water, groundwater, precipitation), and their connectivity to other water bodies, are important parameters controlling isotope ratios (Herzceg et al., 2003; Bocanegra et al., 2013; Wu et al., 2015; Kang et al., 2017, Mass-Dufresne et al., 2021).

The northeastern part of Germany is characterized by a complex system of rivers and lakes. The connectivity of those lakes is an important influence on how ecological and chemical water properties are propagated along the river-lake network. Water isotopes are ideal proxies to identify spatial patterns within these coupled riverine and lacustrine systems and therefore a potential tool to assess lake-to-lake connectivity. So far, available isotope data from this region are limited to large rivers (Reckert et al.,

2017), punctual snapshot data (Richter and Kowski, 1990), or small-scale patterns in local settings (Kuhlemann et al., 2020; Vyse et al., 2020; Kleine et al., 2021).

Here, we provide a comprehensive dataset of water isotopes in lakes along the river-lake system of the northeastern German lowland area (Aichner et al., 2021), which covers the major riverine systems and evaluates the seasonal variability on selected spots, i.e. in lakes which are part of different riverine

branches of the investigated river systems. The two major questions are: a) What is the seasonal variability of water isotope values across the northeastern German lacustrine systems? b) Are there spatial trends that can be unraveled by water isotope data?

For future regional studies, answers to these questions and the assessment if water stable isotopes can be used as accurate proxy for lake-to-lake connectivity, will be valuable in context of the potential

relationship between hydrological, chemical, and biological properties of the studied systems.



## 2 Study site

Northeastern Germany, a section of the glacially formed North German Lowland, is characterized by a complex system of rivers and 1000+ lakes (Fig. 1) with highly variable limnological features such as

size, depth, shape and biogeochemical properties (Ogashawara et al., 2020; 2021). Most of these inland waters are a part of the Elbe watershed, discharging into the North Sea, with the Spree-Dahme system and the Müritz-Havel system as major tributaries. Further to the northeast the Ucker system forms a major cluster of rivers discharging into the Baltic Sea. Smaller natural and artificial channels add to the complexity of this riverine and lacustrine network.

The study area is situated in in the temperate climate zone with an annual precipitation and temperature of approx. 600 mm and 10 °C, respectively (Berlin; DWD, 2021). Isotope values of precipitation range from -84‰ (January) to -49‰ (July) for $\delta^2$H and from -11.6‰ to -6.9‰ for $\delta^{18}$O (OIPC, Online Isotopes in Precipitation Calculator; Bowen and Wilkinsson, 2002; Bowen and Revenaugh, 2003; Stumpp et al., 2014). Due to continental and altitude effects, i.e. isotopic depletion with progressive

transport and rain-out of water vapor, δ-values of precipitation isotopes decrease in the eastern part of Germany from the NW to the SE. This trend is mirrored in groundwater $\delta^2$H values, which decrease in the study area from -60‰ in Lake Müritz to -63‰ in Spree-Dahme rivers (Richter, 1978; Richter and Kowski; 1990).

Lakes sampled for this study are listed in Tables 1 and 2 and can be clustered into nine geographical

groups: I) Lake Müritz and Kölpinsee (Fig 1), which are discharging via the Erpe towards the NW to the Elbe (Fig. 1). II) Deep Müritz-Havel lakes (Fig. 2a), which discharge towards the SE and E before the confluence with the Upper Havel river in Priebertsee: Schwarzer See, Zethner See, Vilzsee, Zotzensee, Labussee, Rätzsee, Canower See, Kleiner Pälitzsee, Großer Pälitzsee. III) The mostly shallow Upper Havel lakes (Fig. 2b), located north of the confluence with the Müritz-Havel water ways:

Zierker See, Useriner See, Großer Labussee, Woblitzsee, Großer Priepertsee. IV) Havel Lakes (Fig. 2d), east of the confluence: Ellbogensee, Ziernsee, Röblinsee and Stolpsee. V) Großer Lychensee, which is connected via the river Woblitz with the Stolpsee. VI) Feldberg Lakes: Feldberger Haussee, Breiter Luzin and Schmaler Luzin.(Fig. 2c), which are connected with each other via surface flow, but only via groundwater flow with the lakes of group V. VII) Lakes connected to the Ucker system, mostly via

smaller creeks and artificial channels (Fig 1 and 2c): Krewitzsee, Mellensee, Wrechener See, Großer See, Große Lanke (southernmost sector of Oberucker See), and Suckower Haussee. VIII) The Spree-Dahme system southeast of Berlin (Fig. 3): Spree, Dämeritzsee, Große Krampe, Dahme, and Müggelsee. IX) Individual lakes Stechlin, and Peetschsee, which are not connected to a major river, but for which connectivity via groundwater flow and small artificial channels is partly still of relevance.

All of these sampled lakes show variable morphometric and ecological characteristics (Tables 1 and 2). The water depths range from 3 m (Zierker See) to 70 m (Lake Stechlin and the volumes from approx. 5.6 million to 99.6 million m$^3$ (same lakes). By lake area, the investigated water bodies range from small ponds such as Peetschsee (19.1 ha) to the largest German lake, the Müritz (10910 ha). Most of the studied northeastern German lakes are mesotrophic or eutrophic according to the classification of

LAWA (LAWA, 2014; Table 1 and 2). Oligotrophic conditions only occur in the Lake Schmaler Luzin. The water residence time (WRT) in these lakes, i.e. the mean time that water spends in a particular lake, has been estimated by means of dividing the lake volume by the flow in or out of the lake (Wetzel, 2001). It ranges from a few weeks to few months in most river-connected lakes, while the Feldberg lakes and Lake Stechlin have longer residence times of ca. 3 to 16 years and >60 years, respectively

(Table 2).

## 3 Methods

### 3.1 Water sampling

Water samples were taken at the shore of 31 lacustrine and riverine spots from 40–60 cm water depth between 8[th]-10[th] March 2020 and 13[h]–19[th] July 2020 (Figs. 1–3; Table 1). At Müggelsee, samples were

taken from a boat pier before the outflow on the northwestern side of the lake, in 2-4 weekly intervals between March 2020 and January 2021. Additionally, more samples were taken at two sites further east on the northern shore of the lake (Fig. 3) on 19[th] June and 13[th] July 2020. Water samples were taken with a pipette and directly transferred in a gas chromatography (GC)-vial, which was closed instantly after sampling and stored cold (6°C) until further processing.





Between March and October 2020, water samples from a subset of 19 lakes (Table 2), were taken in 1-2 months intervals from a boat close to buoys, which were temporarily deployed as part of the project CONNECT at the deepest point of the lake, or (if a site coincided with a water way) near to the outflow of the lake (Ogashawara et al. 2020, 2021): March: 17th–19th. May: 25th–28th. June/July: 29th–2nd. August: 3rd–6th. September: 1st–3rd. October: 5th–8th. On 29th and 30th September, four lakes (Zierker

See, Großer Priepertsee, Ellbogensee, Röblinsee) were sampled at a higher spatial resolution, by taking six or seven surface samples, distributed over the whole lake surface area by positions close to all shores, in addition to the sample at the deepest point and / or center of the lakes (Fig. 2).

A Limnos water sampler (Limnos.pl, Komorów, Poland), capturing 2.5 L volume, was used to obtain water samples from 1 m depth. Samples were transferred into 10 L canisters, closed after sampling and

immediately transported to the lab. Canisters were carefully turned over-top ten times to guarantee constant mixing of samples before samples were transferred into GC-vials with a pipette for isotope analysis. Vials were stored in a cold room (6°C) in the dark until isotope measurement.

## 3.2 Isotope analysis

Water samples were filtered with 0.2 μm cellulose acetate (Faust Lab Science GmbH, Fabrikstraße 17, 79771

Klettgau, Germany) prior to analysis of stable isotopes ($\delta^{18}O$ and $\delta^2H$ values) in the water isotope lab at IGB Berlin, using a L2130-i cavity ring-down spectrometer (Picarro, Santa Clara, CA, USA). Measurements were routinely checked for organic contamination using the Picarro ChemCorrect software.

Isotope values and standard deviations are based on three replicate measurements of each sample, with

additIitional three discarded measurements prior to avoid memory effects. To improve precision, all injections with a water concentration below 17,000 ppm and above 23,000 ppm, and with a standard deviation higher than 400 ppm of the measured water concentration across an injection's averaging window, were excluded.

For instrument calibration 3 laboratory standards for each group of 24 samples were used: L ($\delta^{18}O$ -

17.86‰ and $\delta^2H$ -109.91‰), DEL ($\delta^{18}O$ -10.03‰ and $\delta^2H$ -72.81‰), H ($\delta^{18}O$ 2.95‰ and $\delta^2H$ 0.29‰). A fourth lab standard, M ($\delta^{18}O$ -7.68‰ and $\delta^2H$ -56.70‰), was used as quality and drift control after





every 6 samples. All lab standards were referenced against primary measurement standards: VSMOW2 (Vienna Standard Mean Ocean Water 2), GRESP (Greenland Summit Precipitation, and SLAP2 (Standard Light Antarctic Precipitation 2) from the IAEA (International Atomic Energy Agency,
Vienna International Centre, A-1400 Vienna, Austria).

# 4 Results and discussion

## 4.1 Time series

Isotopes exhibit a clear seasonal trend in the investigated lakes (Figs. 4 and 5). Minimum $\delta^2$H and $\delta^{18}$O values occur at the end of March / early April, while maximum values were measured at the end of
September / early October. In the bi-weekly sampled Müggelsee, the isotope values vary from -53 to -44 ‰ ($\delta^2$H) and from -7.2 to -5.3 ‰ ($\delta^{18}$O), equivalent to seasonal isotope amplitudes of ca 9 ‰ and 1.9 ‰, respectively. The monthly sampled lakes, showed variable seasonal amplitudes (i.e. offsets between October and March), ranging from $\delta^2$H 2–13 ‰ (average: 4 ‰) and $\delta^{18}$O 0.4– 2.6 (average 0.9 ‰).
A seasonal isotopic signal has been observed in many riverine and lacustrine systems (e.g. Dutton et al., 2005; Ogrinc et al., 2011; Halder et al., 2015; Reckerth et al., 2017). Usually, the lake and river isotopes reflect the annual isotope trend of precipitation, but often with significant attenuation of the signal and delay of 1-3 months (Rodgers et al., 2005; Jasechko et al., 2016; Reckerth et al., 2017, Bittar et al., 2017). This is in agreement with data from our study area, where precipitation isotopes reach their
minimum and maximum in Dec/Jan and Jul/Aug, respectively, and exhibit a seasonal variability of ca 35 ‰ ($\delta^2$H) and 4.7 ‰ ($\delta^{18}$O) (OIPC data for Berlin; Bowen and Revenaugh, 2003). The reasons for the time delay between precipitation and river/lake water isotopes, and the smaller seasonal amplitude of the latter, can be attributed to multiple catchment characteristics and processes. Crucial influencing factors on how fast a precipitation isotope signal is transferred into fluvial systems, are the discharge
regime of rivers and the area and topography of their catchment (Sklash et al, 1976; Malozewski et al., 1992; McGuire et al., 2005; Rodgers et al., 2005).

In our study area, the lowest seasonal (Mar-Oct) isotope variability was observed in Lake Stechlin, followed by the Feldberg lakes (Fig. 5), most likely due to their high water residence time and lack of connection to rivers (Table 2). In contrast, the highest amplitude is found in the shallow Zierker See, which also has the lowest volume of all studied lakes. Therefore it is most susceptible to summer evaporation with related isotopic enrichment of the lake water, and also to inflow events, causing more negative values, such as visible in March (Fig. 5). Lake morphology can similarly explain the wide amplitude of Müggelsee (Fig. 4), which is relatively large in area but also among the more shallow of the studied lakes. Data from most Havel lakes, exhibited average seasonal isotope amplitudes (Figs 5 and 7). The Spree-Dahme locations, Kölpinsee and Müritz, and the small Peetschsee, showed relatively large deviations between March and July (Fig. 7). All those lakes are either very shallow or have at least a large area to depth ratio.

These results show that the seasonal isotope amplitude is influenced by multiple characteristics of the studied lakes. Aside morphological parameters such as lake area, depth and volume, hydrological features such as water residence time are of importance.

## 4.2 Spatial patterns

In the four lakes with increased spatial coverage from September samples, little intra-lake variability of isotope values (0.5–1 ‰ for $\delta^2H$ and 0–0.2 ‰ for $\delta^{18}O$) was observed (Fig. 6). In contrast, along the transects, the multiple lakes showed an isotope variability of ca. 25‰ ($\delta^2H$) (Fig. 7).

In general, lakes showed isotopic depletion (values becoming more negative) from the north-west (Müritz) towards the south-east (Spree-Dahme). This follows the similar spatial trend of isotopes in precipitation and groundwater (Richter, 1987; Richter and Kowski, 1990).

In addition, the lakes from the different geographical clusters can be distinguished according to their water isotope signatures: (I) highest values were measured in the two lakes associated with the Müritz-Erpe system (I). The Upper Havel is characterized by lower isotope values than the Müritz-Havel (II and III). An exception are the two lakes Schwarzer See and Zethner See, which show more negative



values than the other Müritz-Havel lakes (Fig. 7). Those form a chain of headwater lakes in the Müritz-Havel system, whose major inflow is north of Zotzensee (Fig. 2a). Schwarzer See is characterized by a

comparably large mean residence time of lake water (Table 2), and is probably under increased influence of isotopically depleted groundwater. The influence of the different isotopic signatures of the two Upper Havel branches is also visible at their confluence in Ellbogensee (Fig. 2a). Here, two samples ELL 2 and ELL 3 (white squares in Fig. 6) taken before the Upper Havel, show approx. 1 ‰ $^2$H enrichment, compared to the samples taken after the confluence. In general, the Havel lakes after the

confluence (IV) carry a mixed isotope signature of the Müritz-Havel and Upper Havel, with stronger influence of the first, depending on season (Fig. 5–7). Isotope depletion was observed towards the Stolpsee, probably due to the influence of the Lychen lakes (V), which discharge into the Upper Havel at Stolpsee via the Woblitz river and which showed the most negative values of all sampled lakes in the Upper Havel system (Figs. 5 and 7).

The Feldberg Lakes (VI) are a group of lakes located approx. 35 km north-east of the Upper Havel system and with insignificant surface connection to the adjacent lake systems. They are characterized by relatively high water residence time and slow surface flow from Feldberger Haussee, to Breiter Luzin, to Schmaler Luzin. $\delta^2$H and $\delta^{18}$O values are higher compared to the Upper Havel and the main branch of the Havel (i.e. after the confluence at Gr. Priepertsee and Ellbogensee), yet these are typical values to

the Müritz-Havel. Among the three lakes, Feldberger Haussee shows highest isotopic values, probably due to its lower water depth and volume (compared to Breiter and Schmaler Luzin) and therefore increased susceptibility to evaporative isotope enrichment.

Variable isotope values are observable in the six sampled lakes associated with the Ucker lake-river system (VII). Complex subsurface and surface flow patterns, mostly via smaller creeks and channels,

their wide geographical distribution and variability in morphometric properties, can explain the isotopic heterogeneity in those lakes. A geographic trend, i.e. lower $\delta^2$H and $\delta^{18}$O values in the Oberuckersee (Große Lanke), compared to the more northern lakes (e.g. Mellensee, Krewitzsee) can be inferred from the data. The most negative isotope values in the study area were measured in samples from the lakes and rivers of the Spree-Dahme system (VIII), located apprx. 100 km south-east of the Müritz-Havel-

Ucker systems.

The final group of lakes is characterized by low surface flow connectivity to riverine systems and other lakes (IX). Those lakes (Lake Stechlin and Peetschsee) show higher isotopic values than adjacent lakes and rivers which can be attributed to evaporation effects. Those in turn are more pronounced in the smaller water bodies, which is visible at the large seasonal isotope amplitude in Peetschsee, compared

to small seasonal variability in the deep and large Lake Stechlin (Fig. 7).

## 5 Conclusions

Stable isotope $\delta^2$H and $\delta^{18}$O values of water samples collected in northeastern German lakes and rivers show clear spatial and seasonal trends. While isotope data exhibit a tendency from relatively high values in the north-west, towards lower values in the south-east, evaporation and groundwater inflow

influence the isotope values on a local scale. In this context, morphometric parameters (water depth, area, volume, and overall shape) and related hydrological properties (water residence time, susceptibility to lake water evaporation and groundwater inflow) are crucial features, with effects on both absolute isotope values and their seasonal amplitude.

The comprehensive dataset of stable water isotopes and lake morphological characteristics described

here can help to set biological and biogeochemical data in context to hydrological processes in the northeastern German riverine and lacustrine systems. The ecological and chemical characteristics of these lake-river systems are driven by a) connectivity between the lakes associated to river systems, in addition to b) local influencing factors which in turn often depend on lake morphology. Water isotopes are able to reflect both those aspects as shown by the different isotope patterns in the different branches

of the Müritz-Havel-Ucker-Spree-Dahme system.

## Author contributions

B.A. designed the isotope study and wrote the manuscript in close cooperation with the CONNECT Project, coordinated by S.A.B., S.W., and performed by C.K., A.J., K.K, I.O., J.C.N, H.-P.G.,F.H., G.S., S.W., and S.A.B.. D.D. analyzed water samples collected by B.A., S.-F.H., K.K., C.K., I.O., A.J.,



S.A.B., S.W., J.C.N., H.-P.G. and F.H.. All co-authors contributed to data evaluation and interpretation, and editing of the manuscript.

## Acknowledgements

We thank Solvig Pinnow, Gregorio Lopez Mazacotte, Bianca Schmid-Paech, Fabian Göppel, Maren
Lentz, Gary Gottschall for help with sampling and sample preparation for isotope analysis. Funding was provided by German Science Foundation (DFG) (project Ai 134/3-1). Part of this work was funded by the Leibniz Association within the Collaborative Excellence programme for the project CONNECT – Connectivity and synchronisation of lake ecosystems in space and time (K45/2017).

## Competing interests

The authors declare no competing interests.

## Data availability

All data in this manuscript are available at PANGAEA: https://doi.org/10.1594/PANGAEA.935633 (Aichner et al., 2021; accessible after acceptance of this manuscript). Temporarily, the dataset can be accessed under this barrier free link:

https://www.pangaea.de/tok/1da04ef79674dd9169df42db2368107323ee4ef4

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



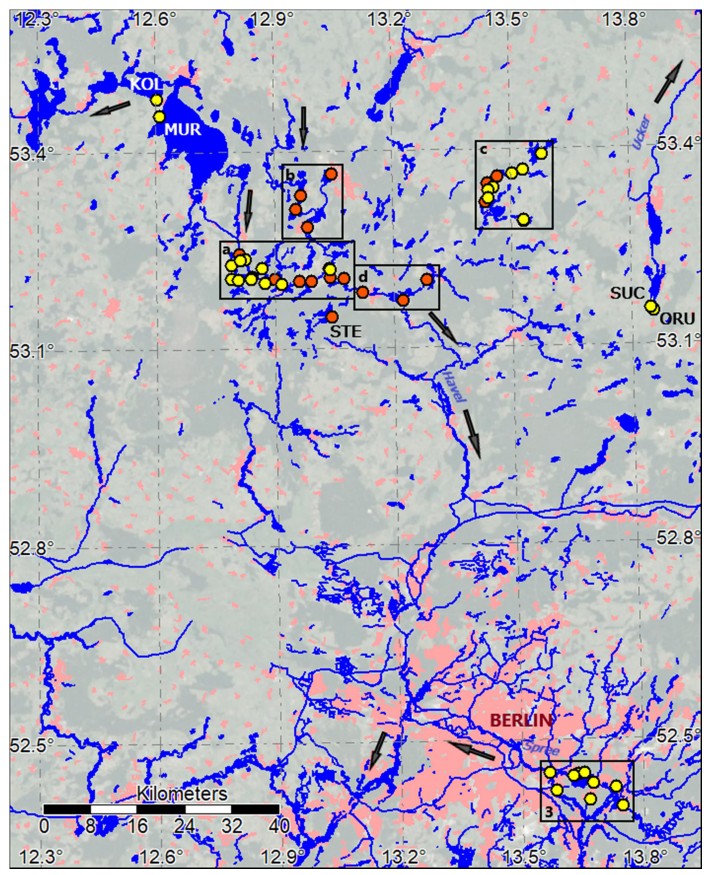

**Figure 1: Sampling locations in the lakes at the shores (yellow) and at the buoys (red). Arrows indicate major flow directions of rivers. Black boxes refer to detailed maps in Fig 2 and 3. Underlying map © Google Maps 2021.**

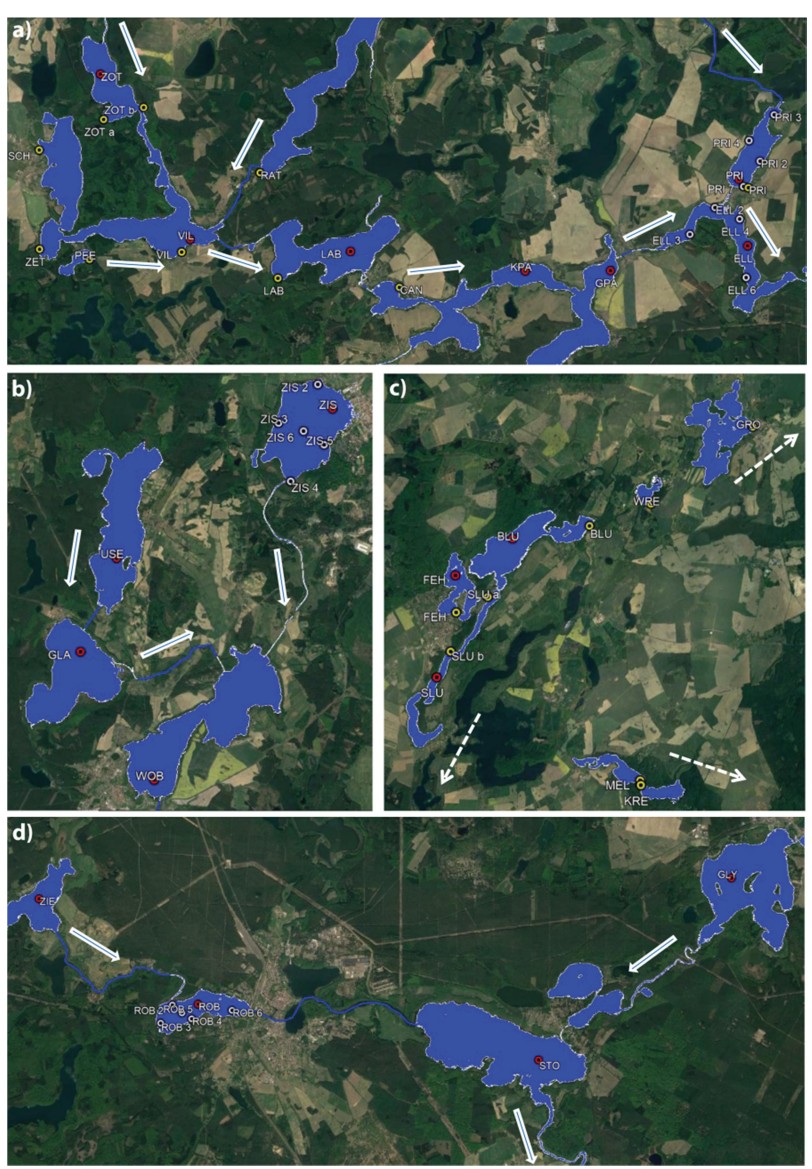

**Figure 2: Sampling locations at (a) the Müritz-Havel, (b) the Upper Havel, (c) in Feldberg Lakes, (d) the Havel main branch. Shore samples taken in March and July 2020 marked yellow. Time series samples from buoys in red. Arrows indicate stream flow direction. Underlying map © Google Earth 2021.**


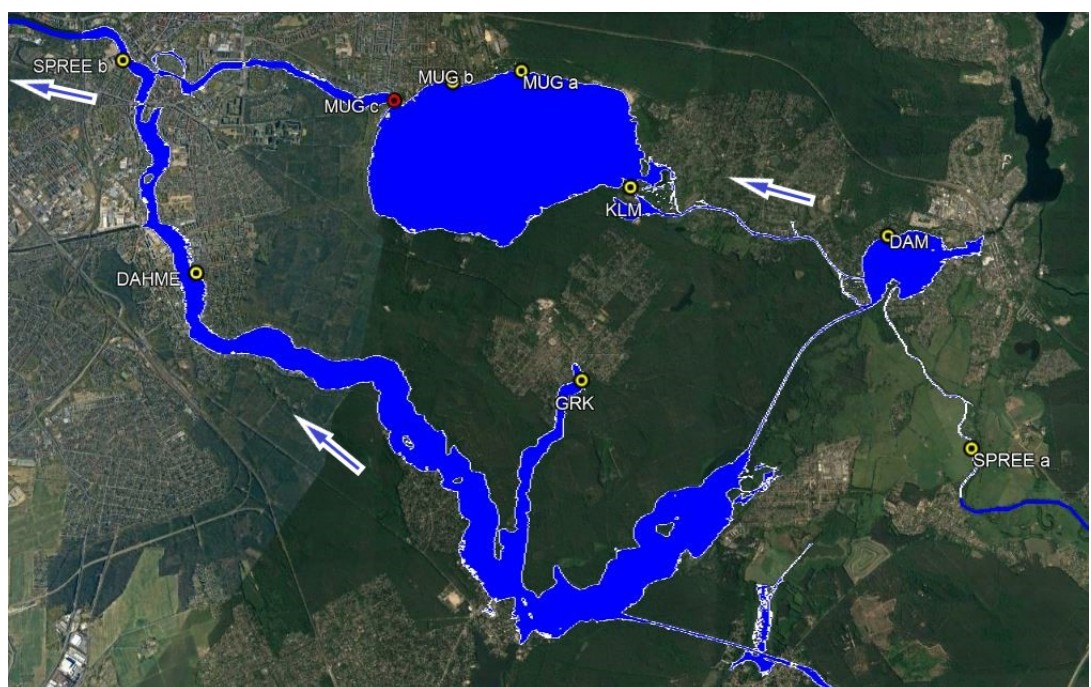

**Figure 3: Sampling locations at the Spree-Dahme. Time series samples from the Müggelsee outflow in red. Arrows indicate stream direction. Underlying map © Google Earth 2021.**



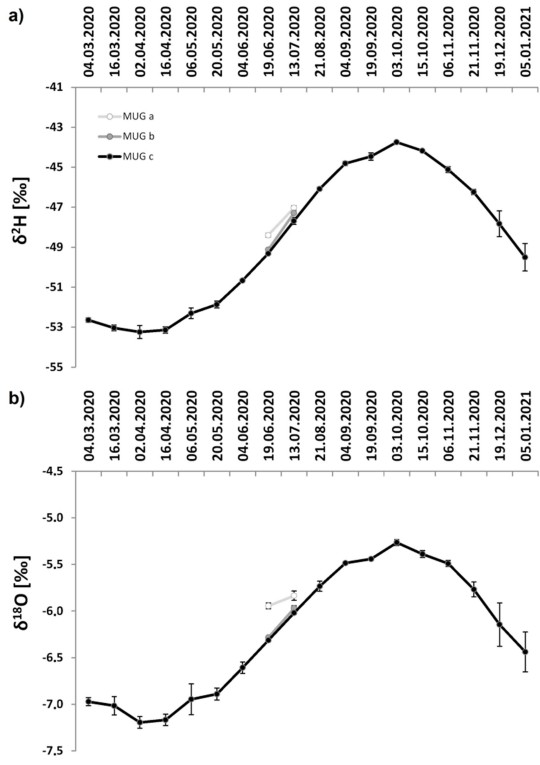


**Figure 4: Timeseries of δ²H and δ¹⁸O values from the samples taken in 2-4 weekly intervals (March 2020 – January 2021) at the outflow of Müggelsee (MUG c: Friedrichshagen harbor) and from additional samples taken on the northern shore (MUG a and b) on 19th June and 13th July.**



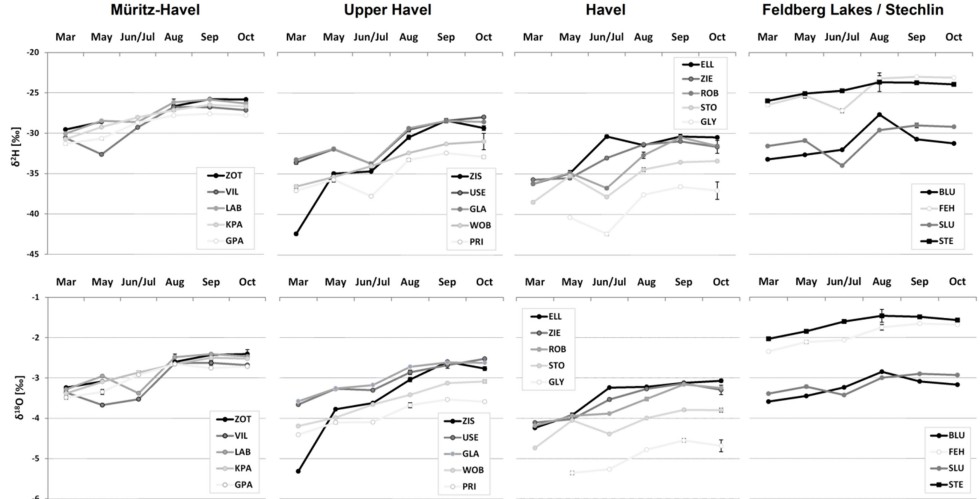

**Figure 5: Time-series of δ²H and δ¹⁸O values of water samples taken from 1 m depth at deepest water depths in lakes connected to the Müritz-Havel and Upper Havel river system and the Feldberg lakes between March and October 2020.**



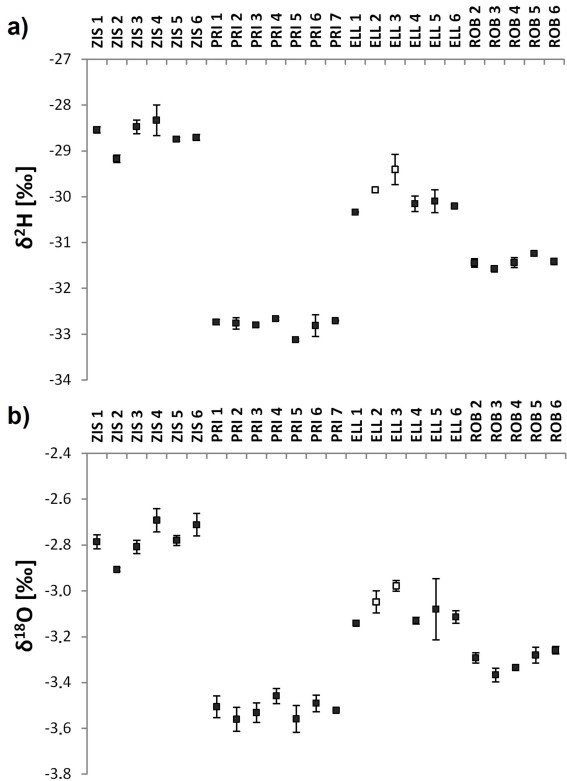

**Figure 6: (a) δ²H and (b) δ¹⁸O values from samples taken at Zierker See (ZIS), Großer Priepertsee (PRI), Ellbogensee (ELL), and**
**Röblinsee (ROB). For locations of samples refer to Table 3 and Fig 2. White squares and black squares for ELL mark samples**
**before and after the confluence with the Upper Havel, respectively.**

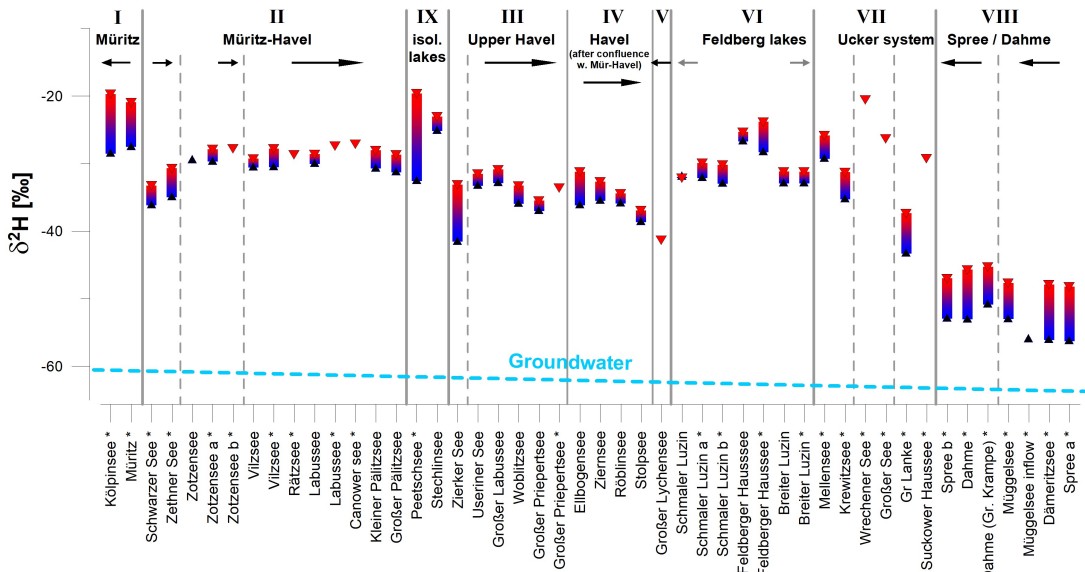

**Figure 7: Seasonal variability of δ²H values between spring (blue) and summer (red) in samples taken at lake shores (\*) and at buoys (1 m depth). Data for spring and summer were taken in March and June / July, respectively. Arrows mark surface (black) and subsurface flow (grey) direction.Triangles mark single data points. Roman numbers indicate geographical clusters as described in section 2. Vertical grey dashed line indicate sub-branches of these clusters (see Fig. 2). Light blue dashed line indicates approximate NW-SE trend in groundwater δ²H values according to Richter, 1987 and Richter and Kowski (1990).**






**Table 1: Parameters of lakes with shore sampling spots. Samples taken from 40-60 cm below water surface. Yellow dots in Fig. 1.**
**Trophy class according to LAWA (2014) with year of most recent evaluation. Data from the respective German states were provided by local authorities (Landesamt für Umwelt Brandenburg, the Ministry of Agriculture and Environment Mecklenburg-Vorpommern, and Berliner Senatsverwaltung).**

| Name | ID | Lat. [°N] | Long. [°E] | Sampl. data I | Samp. date II | Area [ha] | Max depth [m] | Vol. [Mio m³] | Res time [a] | Catchment [km²] | Trophy index / class / year |
|---|---|---|---|---|---|---|---|---|---|---|---|
| Kölpinsee | KOL | 53.480486 | 12.602401 | 10 Mar | 19 Jul | 2009 | 30.0 | 71.8 | 0.930 | 906 | 2.2 / m2 / 2019 |
| Müritz (Outer Müritz) | MUR | 53.454812 | 12.609953 | 10 Mar | 19 Jul | 10201 | 29.2 | 674.9 | 10.1 | 735 | 1.8 / m1 / 2020 |
| Schwarzer See | SCH | 53.228292 | 12.788853 | 10 Mar | 19 Jul | 182 | 34.2 | 22.0 | 23.2 | 15 | 2.4 / m2 / 2020 |
| Zethnersee | ZET | 53.207225 | 12.789568 | 10 Mar | 19 Jul | 38.5 | 6.2 | 1.5 | nd | 20 | 3.1 / e2 / 2017 |
| Peetschsee | PEE | 53.205068 | 12.807274 | 10 Mar | 19 Jul | 18.2 | 10.6 | 0.8 | nd | 1 | 3.1 / e2 / 2016 |
| Zotzensee a | ZOT | 53.234774 | 12.811522 | - | 19 Jul | 149.5 | 21.4 | 10.1 | 0.152 | 116 | 3.3 / e2 /2017 |
| Zotzensee b | ZOT | 53.237293 | 12.825709 | 10 Mar | 19 Jul | | | | | | |
| Vilzsee | VIL | 53.206542 | 12.839634 | 10 Mar | 19 Jul | 200.2 | 21.7 | 15.9 | 0.229 | 147 | 3.2 / e2 / 2017 |
| Rätzsee | RAT | 53.223299 | 12.867060 | - | 19 Jul | 307.4 | 11.5 | 17.9 | 4.064 | 33 | 2.6 / e1 / 2020 |
| Labussee | LAB | 53.200965 | 12.873554 | - | 19 Jul | 258.3 | 26.4 | 18.3 | 0.223 | 210 | 2.9 / e1 / 2014 |
| Canower See | CAN | 53.198937 | 12.916247 | - | 19 Jul | 50.3 | 6.5 | 1.8 | 0.023 | 216 | 2.8 / e1 / 2020 |
| Großer Priepertsee | PRI | 53.219635 | 13.039599 | - | 19 Jul | 105.2 | 26.7 | 11.4 | 0.163 | 412 | 3.2 / e2 / 2015 |
| Schmaler Luzin a | SLU | 53.325860 | 13.441056 | 08 Mar | 16 Jul | 146.2 | 33.5 | 21.2 | 3.359 | 30 | 1.4 / o / 2018 |
| Schmaler Luzin b | SLU | 53.341266 | 13.456269 | 08 Mar | 16 Jul | | | | | | |
| Feldberger Haussee | FEH | 53.336906 | 13.441924 | 08 Mar | 16 Jul | 132.0 | 12.5 | 7.7 | 3.050 | 6 | 2.1 / m2 / 2018 |
| Breiter Luzin | BLU | 53.361765 | 13.502401 | 08 Mar | 16 Jul | 337.3 | 58.3 | 76.9 | 16.254 | 23 | 1.9 / m1 / 2018 |
| Mellensee | MEL | 53.290549 | 13.531495 | 08 Mar | 16 Jul | 73 | 18.3 | 5.4 | 5 | 93 | 2.1 / m2 / 2019 |
| Krewitzsee | KRE | 53.289324 | 13.531882 | 08 Mar | 16 Jul | 57 | 18.8 | 4.1 | 1 | 117 | 2.3 / m2 / 2019 |
| Wrechener See | WRE | 53.368161 | 13.531557 | - | 16 Jul | 44.9 | 4.0 | 0.9 | nd | 12 | 2.6 / e1 / 2019 |
| Großer See | GRO | 53.391470 | 13.580108 | - | 16 Jul | 357 | 20 | 24 | nd | 69 | 3.2 / e2 / 2017 |
| Suckower Haussee | SUC | 53.155456 | 13.847490 | - | 16 Jul | 27 | 6.0 | nd | nd | nd | 2.7 / e1 / 2018 |
| Oberrucker S. (Gr Lanke) | ORU | 53.149541 | 13.854400 | 08 Mar | 16 Jul | 540 | 28.5 | 57 | 4 | 329 | 1.8 / m1 / 2017 |
| Spree b | SPREE | 52.450164 | 13.567648 | 19 Mar | 13 Jul | | | | | | |



| Dahme | DAHME | 52.421870 | 13.585099 | 19 Mar | 13 Jul | | | | | | |
|---|---|---|---|---|---|---|---|---|---|---|---|
| Große Krampe | GRK | 52.407561 | 13.666272 | 19 Mar | 13 Jul | 68 | 5.6 | nd | nd | nd | nd |
| Müggelsee c (outflow) | MUG | 52.444420 | 13.626031 | 19 Mar | 13 Jul | 746 | 7.7 | 36.6 | 0.14 | 7000 | --/ e2 /2014 |
| Müggelsee b | MUG | 52.446808 | 13.638464 | 19 Mar | 13 Jul | | | | | | |
| Müggelsee a | MUG | 52.448213 | 13.653171 | 19 Mar | 13 Jul | | | | | | |
| Müggelsee inflow | KLM | 52.432557 | 13.676486 | 19 Mar | - | | | | | | |
| Dämeritzsee | DAM | 52.425817 | 13.730797 | 19 Mar | 13 Jul | 102.7 | 5.7 | 2.7 | 0.01 | nd | --/ p1 / 2014 |
| Spree a | SPREE | 52.398384 | 13.747033 | 19 Mar | 13 Jul | | | | | | |







**Table 2: Parameters of lakes equipped with buoys. Samples taken from 1 m below water surface. Orange dots in Fig. 1. Trophy class according to LAWA (2014) with year of most recent evaluation. Data from the respective German states were provided by local authorities (Landesamt für Umwelt Brandenburg, the Ministry of Agriculture and Environment Mecklenburg-Vorpommern, and Berliner Senatsverwaltung).**

| Name | ID | Lat. [°N] | Long. [°E] | Trophy index /class / year | Lake area [ha] | Vol. [Mio m³] | Catch ment [km²] | Res. time [a] | Sampl. / max depth [m] | Sampling dates 2020 |
|---|---|---|---|---|---|---|---|---|---|---|
| Zotzensee | ZOT | 53.244500 | 12.810111 | 3.3 / e2 / 2017 | 149.5 | 10.1 | 116 | 0.152 | 20 / 21 | 18 Mar, 26 May, 30 Jun, 03 Aug, 01 Sep, 06 Oct |
| Vilzsee | VIL | 53.209231 | 12.842569 | 3.2 / e2 / 2017 | 200.2 | 15.9 | 147 | 0.229 | 14 / 21 | 18 Mar, 26 May, 30 Jun, 03 Aug, 01 Sep, 06 Oct |
| Labussee | LAB | 53.206569 | 12.899061 | 2.9 / e1 / 2014 | 258.3 | 18.3 | 210 | 0.223 | 17 / 26 | 17 Mar, 26 May, 30 Jun, 03 Aug, 01 Sep, 05 Oct |
| Kl Pälitzsee (E basin) | KPA | 53.202269 | 12.960511 | 2.9 / e1 / 2014 | 132.9 | 7.0 | 224 | 0.075 | 22 / 26 | 19 Mar, 27 May, 02 Jul, 06 Aug, 03 Sep, 05 Oct |
| Gr Pälitzsee (N basin) | GPA | 53.202261 | 12.990519 | 2.6 / e1 / 2014 | 82.4 | 6.5 | 242 | 0.077 | 13 / 15 | 19 Mar, 27 May, 02 Jul, 06 Aug, 03 Sep, 05 Oct |
| Stechlinsee | STE | 53.147460 | 13.031148 | 2.2 / m2 / 2019 | 412 | 99.6 | 26 | 65.0 | 69 / 70 | 17 Mar, 25 May, 01 Jul, 04 Aug, 10 Sep, 06 Oct |
| Zierker See | ZIS | 53.365950 | 13.045639 | 3.7 / p1 / 2018 | 351.0 | 5.7 | 23 | 1.128 | 2 / 3 | 17 Mar, 27 May, 29 Jun, 05 Aug, 03 Sep, 06 Oct |
| Useriner See | USE | 53.332739 | 12.967211 | 2.8 / e1 / 2018 | 376.4 | 17.4 | 161 | 0.725 | 8 / 10 | 17 Mar, 28 May, 29 Jun, 05 Aug, 01 Sep, 06 Oct |
| Großer Labussee | GLA | 53.312300 | 12.954881 | 2.8 / e1 / 2018 | 335.6 | 13.8 | 176 | 0.559 | 8 / 12 | 17 Mar, 27 May, 29 Jun, 03 Aug, 03 Sep, 05 Oct |
| Woblitzsee | WOB | 53.284919 | 12.982481 | 3.3 / e2 / 2015 | 504.5 | 19.8 | 346 | 0.345 | 4 / 7 | 17 Mar, 27 May, 29 Jun, 03 Aug, 03 Sep, 04 Oct |
| Großer Priepertsee | PRI | 53.221431 | 13.036339 | 3.2 / e2 / 2015 | 105.2 | 11.4 | 412 | 0.163 | 16 / 27 | 18 Mar, 26 May, 01 Jul, 06 Aug, 03 Sep, 08 Oct |
| Ellbogensee | ELL | 53.207319 | 13.038950 | 2.9 / e2 / 2015 | 174.0 | 13.4 | 618 | 0.103 | 14 / 18 | 18 Mar, 26 May, 02 Jul, 06 Aug, 03 Sep, 08 Oct |
| Ziernsee | ZIE | 53.205981 | 13.074319 | 2.8 / e1 / 2013 | 112.0 | 6.8 | 634 | 0.052 | 12 / 13 | 18 Mar, 26 May, 02 Jul, 06 Aug, 03 Sep, 08 Oct |
| Röblinsee | ROB | 53.185311 | 13.120869 | 3.0 / e2 / 2017 | 87 | 3.3 | 729 | 7.0 | 7 / 8 | 19 Mar, 25 May, 01 Jul, 04 Aug, 31 Aug, 05 Oct |
| Stolpsee | STO | 53.171819 | 13.221836 | 3.0 / e2 / 2017 | 371 | 24.7 | 1126 | 3.0 | 11 / 13 | 19 Mar, 25 May, 29 Jun, 04 Aug, 31 Aug, 07 Oct |
| Großer Lychensee | GLY | 53.202550 | 13.283633 | 3.3 / e2 /2017 | 282 | 17.6 | 175 | 3.0 | 19 / 19 | --- , 25 May, 29 Jun, 04 Aug, 31 Aug, 07 Oct |
| Schmaler Luzin | SLU | 53.318739 | 13.435600 | 1.4 / o / 2018 | 146.2 | 21.2 | 30 | 3.359 | 32 / 34 | 19 Mar, 28 May, 30 Jun, 05 Aug, 01 Sep, 08 Oct |
| Feldberger Haussee | FEH | 53.347531 | 13.440081 | 2.1 / m2 / 2018 | 132.0 | 7.7 | 6 | 3.050 | 12 / 13 | 19 Mar, 28 May, 30 Jun, 05 Aug, 01 Sep, 08 Oct |
| Breiter Luzin | BLU | 53.358181 | 13.465839 | 1.9 / m1 / 2018 | 337.3 | 76.9 | 23 | 16.254 | 56 / 58 | 19 Mar, 28 May, 30 Jun, 05 Aug, 01 Sep, 08 Oct |




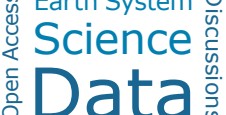

**Table 3: Sampling locations for spatial campaigns in Zierker See (ZIS), Großer Priepertsee (PRI), Ellbogensee (ELL) and Röblinsee (ROB). Samples #1 are identical with buoys as in Table 2 and taken from 1 m below water surface. White dots in Fig. 2.**

| Samping date | Location | Latitude [°N] | Longitude [°E] | Secchi Depth [m] | Water depth at sampl. site [m] |
|---|---|---|---|---|---|
| 29.09.2020 | ZIS 1 | 53.365975° | 13.045650° | 0.45 | 2.7 |
| 29.09.2020 | ZIS 2 | 53.371400° | 13.040172° | 0.35 | 1.3 |
| 29.09.2020 | ZIS 3 | 53.362697° | 13.025783° | 0.4 | 1.3 |
| 29.09.2020 | ZIS 4 | 53.349753° | 13.030467° | 0.45 | 0.7 |
| 29.09.2020 | ZIS 5 | 53.357917° | 13.042639° | 0.4 | 0.8 |
| 29.09.2020 | ZIS 6 | 53.360944° | 13.035014° | 0.5 | 1.95 |
| 29.09.2020 | PRI 1 | 53.221492° | 13.036342° | 1 | 16 |
| 29.09.2020 | PRI 2 | 53.225261° | 13.044003° | 1.1 | > 10 |
| 29.09.2020 | PRI 3 | 53.235164° | 13.049375° | 1 | 2.2 |
| 29.09.2020 | PRI 4 | 53.229672° | 13.040261° | 1 | 5 |
| 29.09.2020 | PRI 5 | 53.224444° | 13.038806° | 1.1 | > 10 |
| 29.09.2020 | PRI 6 | 53.224444° | 13.038806° | 1.1 | 7.2 |
| 29.09.2020 | PRI 7 | 53.219894° | 13.037950° | 1.1 | 9.2 |
| 30.09.2020 | ELL 1 | 53.207383° | 13.038944° | 1.8 | 14.3 |
| 30.09.2020 | ELL 2 | 53.215461° | 13.027700° | 1.85 | 6.3 |
| 30.09.2020 | ELL 3 | 53.209814° | 13.018661° | 1.9 | 10.5 |
| 30.09.2020 | ELL 4 | 53.213000° | 13.036328° | 1.9 | 6.3 |
| 30.09.2020 | ELL 5 | 53.209417° | 13.033500° | 1.75 | 8.2 |
| 30.09.2020 | ELL 6 | 53.200639° | 13.038172° | 1.7 | 6 |
| 30.09.2020 | ROB 1 | 53.185361° | 13.120836° | 1.2 | 5.3 |
| 30.09.2020 | ROB 2 | 53.185428° | 13.113164° | 1.2 | 1.5 |
| 30.09.2020 | ROB 3 | 53.182275° | 13.109422° | 1.2 | 2.8 |
| 30.09.2020 | ROB 4 | 53.182753° | 13.118769° | 1.1 | 5.4 |
| 30.09.2020 | ROB 5 | 53.183917° | 13.115953° | 1.2 | 5.8 |
| 30.09.2020 | ROB 6 | 53.183853° | 13.130997° | 1.2 | 2.5 |