# Peer review of "Spatial and seasonal patterns of water isotopes in northeastern German lakes"

_Earth System Science Data, 2021_

## Author Response (AR1)

**Reviewer #1 (Gabriel Bowen)**

This is a nice new regional lake water isotope dataset from Germany. I have reviewed the dataset on Pangaea, which looks to be in good shape. The accompanying text and figures are appropriate and for the most part of high quality.

Re: we thank the reviewer for the effort of reviewing this data description manuscript (including the underlying data set, deposited on Pangaea) and for the overall positive evaluation. Below are our comments to the (mostly) technical details, as raised by the reviewer.

I have a few technical/presentation comments that should be addressed prior to acceptance:

1. Lines 67-69: please review and follow the guidelines for citing the OIPC:
   https://wateriso.utah.edu/waterisotopes/pages/data_access/oipc_citation.html

Re: we will update the citations to follow the OIPC guidelines.

2. Line 126: you say measurements were "routinely" checked for organic contimination...please clarify, does this mean they were always checked? If not, under what circumstances were they checked and what is the justification for applying those (presumably favorable) results to the rest of the samples?

Re: this was indeed not described sufficiently. We suggest to modify this section as follows "All measurements were post processed with the Picarro ChemCorrect™ software, which compared the measured spectra of the lab standards with the spectra of the samples. If statistical differences (i.e. baseline offset, spectral interference of organic compounds) between the two were too high, a warning flag was assigned and the sample was excluded."

3. Section 3.2: Please discuss and provide information on the analytical uncertainty. The data files report uncertainties (1 sd) for the individual measurements, but the methods section doesn't explain how these were calculated or what they represent. Are the calculated from the replicate injections of each sample? If so, they give a partial measure of uncertainty but do not reflect batch-to-batch or external calibration uncertainty (which ideally would also be known). The methods section mentions a 'check standard' "M" but doesn't present any results or statistics on the reproducibility of the measurements of that standard...reporting this information would be useful in providing a broader metric of uncertainty.

Re: the reported 1 sd uncertainties in the data files are indeed only giving information about variability of δ-values from replicate measurements. We agree that the overall uncertainty of the analytical methodology should be estimated and reported (i.e. including parameters such as errors from calibration with standards, etc, to the overall analytical uncertainty). We

suggest to do this in the method parts of the data description manuscript, but information could also be added to the abstracts of the Pangaea data sets.

With respect to the "M" standard we suggest to rephrase this section to clarify: "A fourth lab standard, M ($\delta_{18}O$ -7.68‰ and $\delta_2H$ -56.70‰), was used as quality and drift control after every 6 samples, i.e. serving as lab-internal control and not being used for calibration. Mean values calculated from all measured M lab standards were -7.68‰ for $\delta_{18}O$ with a SD of 0.10‰ and -55.93‰ for $\delta_2H$ with a SD of 0.31‰."

4. Lines 174-175: Please review the wording of this sentence, there is a word missing or out of place here.

Re: we will rephrase this sentence.

5. Figure 1: I would be helpful to provide either lat/lon ticks on the axes of this map or an inset showing the broader geographic context (e.g., outline of Germany or N Europe showing the study location) to help orient the reader.

Re: lat/lon ticks are included in the map

**Reviewer #2**

Aichner et al (2021) present a dataset of O and H stable isotope measurements in samples collected from lakes in NE Germany in 2020. Such data are much needed if one is to understand (eco)hydrological processes in the water cycle.

However, the dataset offered here by the authors lacks in several aspects, making difficult to be used as potentially intended. As provider and user of similar data I find this dataset lacking several important assets which I highlight below.

Re: we thank the review for the effort of reviewing this manuscript and the positive evaluation about the principal value of the data set. Below we reply to some specific concerns.

1. The variability of isotopologues in lake waters, similar to that in precipitation for example, follow a clear annual cycle. In order to understand these dynamics, at least one full yearly cycle needs to be covered by sampling. The authors collected waters mostly during the (extended) warm season, with several cases in which only two samples/year (warm season) were collected. In the absence of winter samples, it is

impossible for the data to "give information about the seasonal isotope amplitude (page1, line 26). I am not sure I understand how this can be done with the present dataset. Further, in the absence of samples collected in winter (except for one case), it is virtually impossible to understand 1) the links between stable isotopes in lake waters on one hand, and stable isotopes in precipitation and weather/climate on the other hand, 2) recharge patterns and their timing.

Re: the measured samples give a good estimate about the seasonal isotope variability in most of the studied lakes. This was considered when conceptualizing the sampling strategy, i.e. we took care that the minimum and maximum of the annual cycle of δ-values will be tracked.

The lowest/highest isotope values in these riverine lakes are reached in late March/early April and early October, respectively. This nicely comes out in the data from lake Müggelsee, which was sampled over a longer period and in higher time resolution (Figure 4).

Therefore, the data derived from the applied sampling strategy in nineteen of the studied lakes (data shown in Figure 5), which were sampled among others in mid March and early October (Table), provide a very good estimate of the seasonal isotope amplitude.

Indeed, the additional shore samples (taken partially from similar lakes, and partially from additional lakes), cannot provide an estimate about the full seasonal amplitude. However, the applied strategy here, i.e. sampling in mid/late March and mid of July, frames the growing period of vegetation. Especially aquatic vegetation relies on source water from such lakes, therefore the "ecological isotope amplitude" constrains the possible range of an isotopic signal which is incorporated into the plant biomass during growth. This is very useful for ecological studies, which depend on that parameter, from either the studied lakes or comparable ones in the closer or wider region.

=> currently Figure 7 only illustrates the "ecological amplitude" i.e. the March-July isotope offset. We suggest to also plot the "full seasonal isotope amplitude", for the nineteen lakes mentioned above plus Müggelsee, from which these data are abundant. Following this, October isotope values were additionally plotted into Figure 9. In relation to this, also the scientific values of those two amplitudes will be emphasized.

Concerning relationship to weather/climate and precipitation isotopes, please see below.

2. The stable isotope data is not accompanied by any physical, chemical or hydrologic data so understanding their temporal and spatial dynamics is almost impossible to understand (are these caused by hydrological processes? climatic ones?). For example, seasonal amplitude (see my comment above on seasonality) "can be attributed to multiple catchment characteristics and processes" (page 6, lines 157-158). Of course they can, these are the factors affecting the O and H stable isotope values in all lakes across the Globe, but nothing can be said on the investigated lakes here.

In my view, the sentence quoted above summarizes the maximum value that can be obtained from the dataset as it is now. I don't know if more data is available, but if not, I suggest the authors write a scientific article analyzing their data and append the data to the article. It would better served the wider scientific community.

Re: a range of physical, chemical and hydrological data is available from most of the studied lakes.

Many morphological parameters (depth, area, volume, water residence time, catchment area) are listed in Table 1 and 2 of the data description manuscript. Also, trophic classification to enable ecological contextualization, is listed there.

=> It would be good to know, which additional parameters (i.e. not listed in Table 1 and 2) the reviewer precisely is referring to. For the revised version further columns with water temperature, pH, and O2-saturation were added to the Müggelsee-timeseries (in the Pangaea dataset and plotted in Fig. 4). Further water temperatures were to the other March-October lake timeseries (in the Pangaea dataset, not shown in Figures because trends are similar in those connected lake systems).